# Multi-view X-ray Image Synthesis with Multiple Domain Disentanglement from CT Scans

## ABSTRACT

X-ray images play a vital role in the intraoperative processes due to their high resolution and fast imaging speed and greatly promote the subsequent segmentation, registration and reconstruction. However, over-dosed X-rays superimpose potential risks to human health to some extent. Data-driven algorithms from volume scans to X-ray images are restricted by the scarcity of paired X-ray and volume data. Existing methods are mainly realized by modelling the whole X-ray imaging procedure. In this study, we propose a learning-based approach termed CT2X-GAN to synthesize the X-ray images in an end-to-end manner using the content and style disentanglement from three different image domains. Our method decouples the anatomical structure information from CT scans and style information from unpaired real X-ray images/ digital reconstructed radiography (DRR) images via a series of decoupling encoders. Additionally, we introduce a novel consistency regularization term to improve the stylistic resemblance between synthesized X-ray images and real X-ray images. Meanwhile, we also impose a supervised process by computing the similarity of computed real DRR and synthesized DRR images. We further develop a pose attention module to fully strengthen the comprehensive information in the decoupled content code from CT scans, facilitating high-quality multi-view image synthesis in the lower 2D space. Extensive experiments were conducted on the publicly available CT-Spine1K dataset and achieved 97.8350, 0.0842 and 3.0938 in terms of FID, KID and defined user-scored X-ray similarity, respectively. In comparison with 3D-aware methods ($\pi$-GAN, EG3D), CT2X-GAN is superior in improving the synthesis quality and realistic to the real X-ray images.

## CCS CONCEPTS

• **Computing methodologies** → **Appearance and texture representations**.

## KEYWORDS

Generative Adversarial Networks, Image Synthesis, X-ray, Style Disentanglement, Multi-domains

## 1 INTRODUCTION

Due to the high resolution and rapid imaging speed, X-ray images are commonly utilized for visualizing the internal anatomical structures of the human body and are regarded as the golden standard for disease diagnosis and treatment. In the X-ray imaging processes, according to the different attenuation coefficients, organs and tissues present various grey distributions[34, 37, 38]. However, the commonly used mono-plane imaging system can only capture the X-ray image from a single view each time. Due to the projecting principle, much spatial information has been lost in the X-ray images and repeated or multiple imaging procedures are further needed to visualize the rich and comprehensive information of human structures[29, 32]. In such processes, excessive doses of X-rays pose unavoidable potential risks to the human body. Besides these, X-ray images play a vital role in the segmentation[6, 45], registration[43, 50], reconstruction [15, 22, 24] and synthesis [26]. Along with the explosive development of deep learning, large amounts of X-ray image databases are needed to extensively improve the performance of the above researches[4, 42]. Hence, synthesizing X-ray images from volume data is desperately needed. In the synthesizing process of X-ray images, attenuation coeffi-

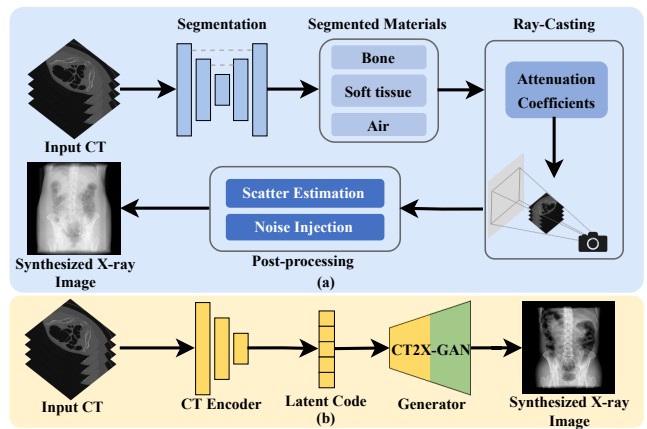

**Figure 1: Illustration of the difference between the traditional and proposed pipeline of X-ray image synthesis from 3D volume scans.**

cients of different parts and interaction between X-rays and structures both need to be accurately modelled. Digital reconstructed radiography (DRR) is a common solution to tackle the problem by computing the attenuation coefficients and modifying the ray-casting approach[13]. However, DRR is highly dependent on the accuracy of each structure segmentation which reduces the generalization ability. Besides, particle numerical models are also essential to simulate the interaction between X-rays and structures which further increases the difficulty and complexity of imaging

procedure[39, 41]. Hence, improving the reality of synthesized X-ray images still faces great challenges.

In this study, we propose a novel and practical task termed CT-to-X-ray (CT2X) synthesis, aiming to generate high-quality and realistic X-ray images from multiple view angles. The goal is to learn the intricate mapping relationship from CT scans to X-ray images. As depicted in Figure 1, the CT scan is directly fed into the generator to synthesize X-ray images, instead of modelling the complicated processes including segmentation, ray-casting, and post-processing in traditional methods. To reach this purpose, CT2X-GAN is realized from the perspectives of style decoupling and pose perception.

To achieve style decoupling, we propose a multiple domain style decoupling encoder, which is capable of decoupling anatomy structure and texture style from 2D X-ray images. To guide the style decoupling encoder in learning content commonalities and style differences across multiple domains, we introduce a novel consistency regularization term to constrain the training process. Additionally, zero loss is introduced to minimize the similarity of style features among reference images from different domains, ensuring the style decoupling encoder focuses on domain-independent style features. For pose perception, we propose a CT encoder incorporating a pose attention module (PAM), enabling the multi-view consistency of X-ray images from multiple view angles using 2D networks.

In summary, our contributions can be summarized as fourfold:

- A novel pipeline is proposed for end-to-end X-ray image synthesis from CT scans without modelling the whole X-ray imaging procedure.
- A style decoupling encoder is introduced to extract style features from real X-ray images, relieving the difficulty of collecting paired CT scans and X-ray images.
- A novel regularization method is developed by utilizing consistency and zero loss to improve style accuracy, improving the structural consistency in multi-views and style decoupling capabilities.
- A PAM is designed to calculate attention based on the projection of the target pose, enabling the network to improve the perception ability of structural content at multiple view angles.

## 2 RELATED WORKS

### 2.1 Traditional X-ray Image Synthesis

The synthesis of X-ray images from volume data aims to obtain realistic X-ray images with consistent structures and styles, thereby reducing patient exposure to X-ray radiation. Mainstream methods are reached by modelling X-ray imaging procedures to generate DRRs [13]. Early research primarily employed ray-casting for synthesizing X-ray images [25]. More sophisticated methods explored the Monte Carlo (MC) [2, 11] method to accurately simulate the interaction between X-ray particles and tissue organs. Li et al. [27] devised an adaptive MC volume Rendering algorithm, partitioning the volume into multiple sub-domains for sampling, thus accelerating the MC simulation.

### 2.2 Learning-based X-ray Image synthesis

Recent advancements in deep learning have significantly promoted X-ray image synthesis performance. Methods such as DeepDRR [30] utilize convolutional neural networks for tissue segmentation in CT scans, alongside ray-casting and MC methods to compute tissue absorption rates. Gopalakrishnan et al. [40] accelerated the synthesis of DRRs by reformulating the ray-casting algorithm as a series of vectorized tensor operations, facilitating DRRs interoperable with gradient-based optimization and deep learning frameworks. It is noteworthy that existing methods heavily rely on the segmentation results from the CT scans. Fixed absorption coefficients are assigned to different structures to calculate X-ray beam attenuation rates. Recent advancements in X-ray projections synthesis have revealed that deep learning models trained on simulated DRRs struggle to generalize to actual X-ray images [30, 44]. However, recent progress in image translation has demonstrated that utilizing condition features extracted from representation learning models can markedly reduce the disparity between synthesized DRRs and real X-ray images [13]. Inspired by this, our model diverges from traditional approaches to modelling of imaging procedures. Instead, it employs style-based generative models, offering a robust approach for disentangled synthesis, thus improving the performance and image quality of X-ray synthesis.

### 2.3 GAN-based Image synthesis

In recent years, generative adversarial networks (GANs) have achieved remarkable success in image synthesis [46], image translation [18], and image editing [1]. Building upon progressive GAN [19], Karras et al. proposed StyleGAN [20], which enhanced the quality of the generated image and allowed the network to decouple different features, providing a more controllable synthesis strategy. StyleGAN2 [21] redesigned the instance normalization scheme to remove the water droplet-like artefacts existing in StyleGAN, leading to higher-quality outputs. Some methods have attempted to control generated images by exploring the latent space of GANs [10, 33, 36]. Despite their remarkable process, most deviations of GANs still concentrate on data augmentation in medical imaging, with limited research conducted on synthesizing cross-dimension medical images [17]. One of the primary challenges inhibiting GANs from the CT2X task is the synthesis of multi-view results, as GANs have limited awareness of pose information [17, 28].

Recently, there has been a trend to incorporate 3D representations into GANs, enabling them to capture pose information in 3D space [28]. Such approaches, known as 3D-aware GANs, facilitate multi-view image synthesis. Methods that introduced explicit 3D representations, such as tree-GAN [48], MeshGAN [9], and SDF-StyleGAN [49], are capable of synthesizing the 3D structure of the target object, which allows them to explicitly define poses in 3D space. However, the use of explicit 3D representations has constrained their resolution. $\pi$-GAN [8] and EG3D [7] introduced radiance fields to endow networks with the capability of pose awareness. Nonetheless, the high computational cost of stochastic sampling required for radiance fields introduces training complexity and may lead to noise [23]. In contrast, our approach aims to improve the capability of the network to perceive spatial poses by involving the projections at the target view angles.

# 3 METHODS

## 3.1 Problem Definition

The purpose of X-ray image synthesis is to predict the X-ray image at any arbitrary view angle from the 3D CT scan. Let us denote the input CT scan with $V \in \mathbb{R}^{H \times W \times D}$, the X-ray image for styles with $I_X \in \mathbb{R}^{H \times W}$, the referenced DRR for structures with $I_{DRR} \in \mathbb{R}^{H \times W}$. $H$, $W$ and $D$ represent the height, width and depth, respectively. $p \in \mathbb{R}^{1 \times 25}$ represents the camera pose of the target view angle. Hence, A mapping function from CT scan to X-ray images can be expressed as follows:

$$f(V, p, I_X) \rightarrow \hat{I}_{X,p} \in \mathbb{R}^{H \times W} \tag{1}$$

In this study, two main challenges in X-ray image synthesis will be tackled: (1) How to conduct end-to-end synthesis using unpaired CT and X-ray data? (2) How to realize X-ray image synthesis from multiple view angles? To address them, we train the network using images from three domains, including CT scans, X-ray images, and DRR images. In Section 3.3, we utilize the CT encoder to extract anatomical information from CT data, while the disentangled encoder processes X-ray images to extract X-ray style features. Our generator seamlessly integrates unpaired information for synthesis. Moreover, we leverage the style features extracted from DRRs to produce style reconstructed syntheses and hence provide supervision. The integration of information from three domains enables the tackling of end-to-end training with unpaired data. Additionally, a consistency regularization is employed in Section 3.4 to further constrain the training of the decoupling encoder, ensuring comprehensive extraction of domain-specific style information. A PAM is introduced to focus the content code on the target view angle, facilitating multi-view synthesis, as depicted in section 3.5. In Section 3.6, we employ a pose-aware adversarial training strategy by feeding the corresponding DRRs into the discriminator, further improving the quality of multi-view synthesis.

## 3.2 Overview of CT2X-GAN

CT2X-GAN synthesizes multi-view X-ray images by incorporating images from three domains: CT scans $V$, X-ray images $I_X$, and DRRs $I_{DRR}$ as inputs during training, as depicted in Figure 2. The CT scan is employed to compute the CT content code $f_c^{CT}$ through a CT encoder $E_{CT}$, whereas the X-ray image is sent to a style decoupling encoder, denoted as $E_{sty}$, to extract the X-ray style code $f_{sty}^X$ and X-ray content code $f_c^X$. The CT content code and X-ray style code are then fed into distinct layers of the generator $G$ to conduct the X-ray image synthesis $\hat{I}_X$. The synthesized X-ray results lack corresponding ground truth (GT). To provide supervision to the network training, we utilize the DeepDRR [30] framework to generate target view angle DRRs $I_{DRR}$ from CT scans, which also serve as the GT DRRs. These DRR images are then sent into the style decoupling encoder to obtain DRR style code $f_{sty}^{DRR}$ and DRR content code $f_c^{DRR}$. Forwarding the DRR style code and the CT content code into the generator produces a DRR stylized image $\hat{I}_{DRR}$. Reconstruction loss can then be computed between the reconstructed and the GT DRR for supervision. A consistency regularization is next introduced to constrain the training and improve the disentanglement by minimizing the discrepancy between the style code and content code between the real and synthesized images. Furthermore, to improve the perception of CT content code to the information of the target pose $p$, a PAM is utilized to modify the CT content code $f_c^{w^+}$ by the maximum intensity projection at the target pose.

## 3.3 Style Decoupling Encoder

To address the challenge of limited paired CT and X-ray data, we propose a style decoupling encoder $E_{sty}$ to separate images into content and style code. This allows the generator to control the style feature in the synthesized images. Specifically, for X-ray reference images $I_X$, the style decoupling encoder consists of two branches, including an X-ray style branch and a content branch. The two branches separate the X-ray image into style code $f_{sty}^X$ and content code $f_c^X$ as follows:

$$f_{sty}^X = E_{sty}^{s_X}(I_X) \tag{2}$$

$$f_c^X = E_{sty}^c(I_X) \tag{3}$$

where $E_{sty}^{s_X}$ and $E_{sty}^c$ denote the X-ray style branch and content branch of the style decoupling encoder, respectively.

The X-ray style code $f_{sty}^X$ describes the low-level style and intensity information of the X-ray image, while the modified content code $f_c^{w^+}$ contains the high-level anatomical information. The generator $G$ synthesizes the X-ray images by combining the X-ray style code with the content code as follows:

$$\hat{I}_X = G(f_c^{w^+}, f_s^X) \tag{4}$$

To provide auxiliary supervision to the network training, we employ the DRRs $I_{DRR}$ computed from the input CT scan as the GT. The reconstructed images $\hat{I}_{DRR}$ synthesized by the style code from such DRRs and the content code from CT scans should be consistent with $I_{DRR}$. To realize this, we introduce a DRR style branch into the style decoupling encoder to be distinct from the X-ray style branch, thus enabling the style decoupling encoder. The style decoupling encoder is utilized to extract style code $f_{sty}^{DRR}$ and content code $f_c^{DRR}$ from DRRs as follows:

$$f_{sty}^{DRR} = E_{sty}^{s_{DRR}}(I_{DRR}) \tag{5}$$

$$f_c^{DRR} = E_{sty}^c(I_{DRR}) \tag{6}$$

where $E_{sty}^{s_{DRR}}$ denotes the DRR style branch of the style decoupling encoder.

The reconstructed DRRs $\hat{I}_{DRR}$ can be obtained by feeding the DRR style code and the modified CT content code into the generator:

$$\hat{I}_{DRR} = G(f_c^{w^+}, f_s^{DRR}) \tag{7}$$

To improve the preservation of anatomical structures and improve the quality of image synthesis, we compute the supervised reconstruction loss between the reconstructed DRRs $\hat{I}_{DRR}$ and GT DRRs $I_{DRR}$ as follows:

$$\mathcal{L}_{rec} = \lambda_{mae} \mathcal{L}_{mae}(I_{DRR}, \hat{I}_{DRR}) + \lambda_{lpips} \mathcal{L}_{lpips}(I_{DRR}, \hat{I}_{DRR}) \tag{8}$$

Here, $\mathcal{L}_{mae}$ is the mean absolute error (mae) loss, $\mathcal{L}_{lpips}$ denotes the learned perceptual image patch similarity (lpips) [47]. $\lambda_{mae}$ and

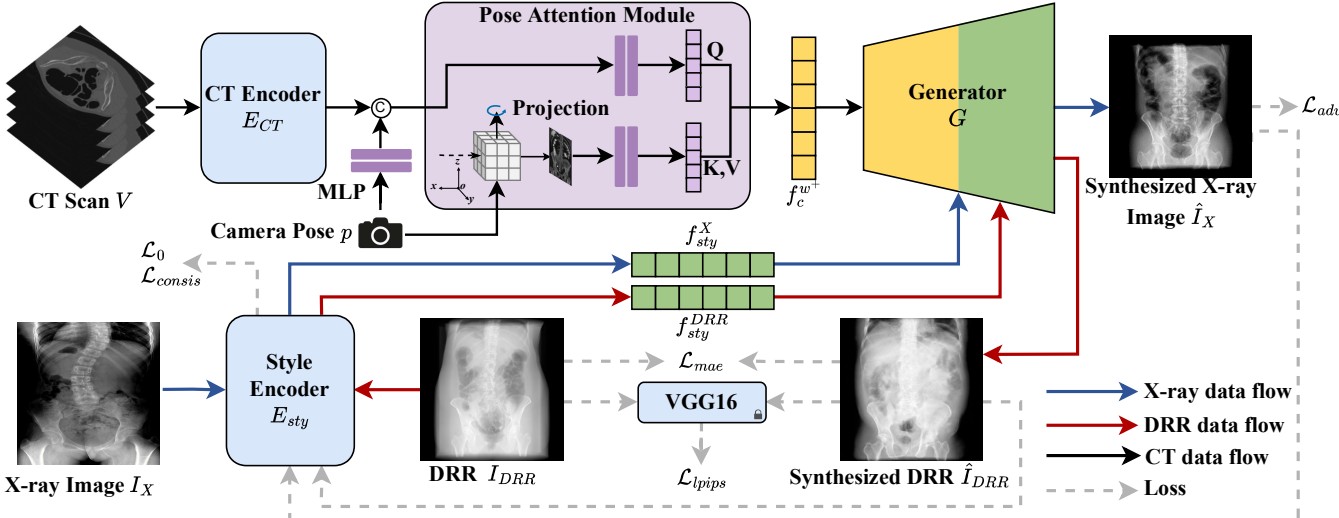

**Figure 2: An illustration of the proposed CT2X-GAN. A CT encoder $E_{CT}$ extracts the content code from the input CT scan $V$. The style decoupling encoder $E_{sty}$ extracts a style code $f_{sty}^X$ from the input X-ray image $I_X$. The generator $G$ incorporates both the style code and content code to generate the X-ray synthesis result $\hat{I}_X$. We also employ the style decoupling encoder to extract a DRR style code $f_{sty}^{DRR}$ from the DRR and use it to synthesize a stylized reconstructed DRR $\hat{I}_{DRR}$, providing auxiliary constraint to the training. A consistency regularization term is calculated to improve domain-specific style extraction. Additionally, we employ a pose attention module to accentuate features with the target view angle based on the corresponding projection.**

$\lambda_{lpips}$ are the hyperparameters controlling the weight of the loss items.

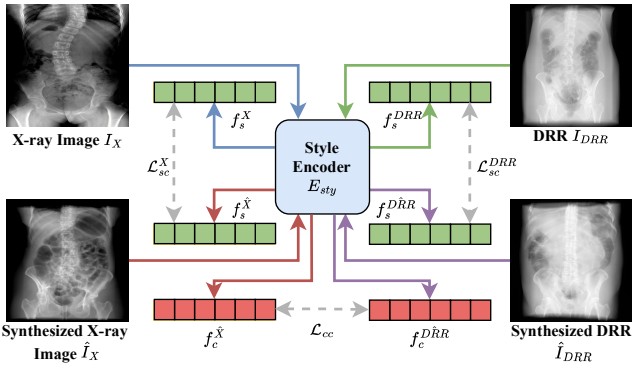

**Figure 3: An illustration of consistency regularization. $f_s^X$ and $f_s^{DRR}$ represent the style code extracted from X-ray and DRR images. $f_s^{\hat{X}}$ and $f_s^{D\hat{R}R}$ are the style features of synthesised results using X-ray and DRR, respectively. $f_c^{\hat{X}}$ and $f_c^{D\hat{R}R}$ share the same contents.**

## 3.4 Consistency Regularization

In decoupling the style information, it is challenging and unstable to supervise and train the decoupling encoder due to the absence of paired X-ray images. Hence, to extract domain-specific style features, we propose a novel consistency regularization approach by combining disentanglement learning and consistency to constrain the decoupling encoder. To ensure that the synthesized images obtain style information only from the style code while preserving the original structure, the decoupling encoder should thoroughly disentangle style and content information from the style image as much as possible. This implies that style branches $E_{sty}^{s_*}$ should extract all essential style information, while structural information can only be extracted by content branch $E_{sty}^c$. We compel the network to extract consistent style codes from the generated image $\hat{I}_*$ and input style image $I_*$, simultaneously ensuring consistency of content codes $f_c^{\hat{X}}$ and $f_c^{D\hat{R}R}$. Therefore, we formulate the following consistency constraints:

$$\mathcal{L}_{consis} = \lambda_{cc}\mathcal{L}_{cc} + \lambda_{sc}\mathcal{L}_{sc} \tag{9}$$

$$\mathcal{L}_{cc} = ||E_{sty}^c(\hat{I}_X) - E_{sty}^c(\hat{I}_{DRR})||_2 \tag{10}$$

$$\mathcal{L}_{sc} = ||E_{sty}^{s_X}(I_X) - E_{sty}^{s_X}(\hat{I}_X)||_2 \\ + ||E_{sty}^{s_{DRR}}(I_{DRR}) - E_{sty}^{s_{DRR}}(\hat{I}_{DRR})||_2 \tag{11}$$

where $\mathcal{L}_{cc}$ describes the content consistency and $\mathcal{L}_{sc}$ represents style consistency. $\lambda_{cc}$ and $\lambda_{sc}$ control the weight of the regularization term.

For the purpose of ensuring that the style encoders can thoroughly disentangle domain-specific style information, we incorporate zero loss [3] as an auxiliary constraint during training. As discussed in Section 3.3, we have designed two domain-specific style branches within the decoupling encoder: one for capturing features of X-ray style and the other for capturing DRR features. Ideally, the X-ray branch should extract appearance features

specific to the real X-ray images, while the DRR branch should extract appearance features specific to the DRRs. Therefore, when a style reference image is inputted into the corresponding branch that does not pertain to the domain, the objective is to minimize the errors extracted by the style encoder.

$$\mathcal{L}_0 = ||E_{sty}^{s_X}(I_{DRR})||_1 + ||E_{sty}^{s_{DRR}}(I_X)||_1 \tag{12}$$

To ensure that each branch extracts style-specific features effectively, the extracted style information should be zero when the input style image belongs to a different domain.

## 3.5 Pose Attention Module

While CT scans contain rich spatial information in 3D space, encoding it into latent space can incur notable information loss. Fully exploiting the 3D inherent in CT scans can lead to a more accurate anatomical structure in the synthesis. Therefore, we introduce a PAM, which is capable of accentuating distribution information of a specific view angle in the CT code $f_c^{CT}$, thereby obtaining an modified content code $f_c^{w^+}$.

Given that the CT scan encompasses comprehensive anatomical information about the human body, the information within the generated X-ray image at a specific view angle should encompass the position, tissue size and morphology of anatomical structures that appeared in the CT scans. Thus, we can project the CT scans to the imaging plane of a target view angle and utilize the projection as auxiliary information to modify the content code through attention mechanisms. Initially, we encode the input CT scan $V$ into a CT code $f_c^{CT}$ using an encoder. Next, the camera parameters $p$ are fed into a multi-layer perceptron (MLP) and concatenated with $f_c^{CT}$ as conditional input. Subsequently, the conditional information is merged with the CT code using an MLP and denoted as **Q**. We then rotate the input volume $V$ in 3D space to align with the target pose $p$ for maximum intensity projection. The resulting projection is encoded via an MLP and utilized as **K** and **V**. Thus, the PAM can be represented as follows:

$$PAM = Softmax(\frac{\mathbf{Q} \cdot \mathbf{K}^T}{\tau})\mathbf{V} \tag{13}$$

$$\mathbf{Q} = Concat(E_{CT}(V), MLP(p)) \tag{14}$$

$$\mathbf{K} = \mathbf{V} = MLP(Proj(Rot(V, p))) \tag{15}$$

where $Concat$ refers to the concatenation, $Rot$ denotes the 3D rotation operation, $Proj$ represents the maximum intensity projection and $\tau$ is a small-valued constant preventing the extreme magnitude.

Overall, our proposed PAM enhances the generator capability of robust multi-view synthesis. By intensifying specific pose features, the PAM facilitates high-quality multi-view synthesis, empowering the 2D network with enhanced 3D perceptual capabilities without the need for 3D representations in the network.

## 3.6 Loss Function

**Pose-aware adversarial training**. To further promote pose consistency between the pose of generated X-ray image $\hat{I}_X$ and the target pose $p$, we integrate pose information into the adversarial training process. This integration enables the model to learn both the

anatomical features and positional characteristics essential for precise multi-view image synthesis. Conditioning the synthesis process on the specified camera pose enables the model to effectively capture geometric details and orientation of the target anatomy, yielding synthesized images with increased fidelity and realism. The overall quality of the generated X-ray images can be improved, facilitating more accurate and clinically relevant interpretations. We formulate the pose-aware adversarial loss as follows:

$$\mathcal{L}_{adv}^{gan} = \mathbb{E}_{v \sim V, x \sim I_X, \phi \sim p}[-D(G(v, x, \phi))] \tag{16}$$

$$\mathcal{L}_{adv}^{dis} = \mathbb{E}_{v \sim V, x \sim I_X, \phi \sim p}[D(G(v, x, \phi))] \\ + \mathbb{E}_{x \sim I_{DRR}}[-D(x) + \lambda_{R_1}|\nabla D(x)|^2] \tag{17}$$

where $V$ is the distribution of CT scans. $I_X$ and $I_{DRR}$ are the distributions of X-ray and DRR images. During training, the synthesis requires a pose $\phi$ sampling from the pose distribution $p$. $\nabla(\cdot)$ is the $R_1$ loss following [31] to stable the training process. $\lambda_{R_1}$ is the balancing weights.

**Final loss**. The final losses for training the generator and the discriminator are then defined as:

$$\mathcal{L}_{total}^{gan} = \lambda_{adv}\mathcal{L}_{adv}^{gan} + \mathcal{L}_{rec} + \mathcal{L}_{consis} + \lambda_0\mathcal{L}_0 \tag{18}$$

$$\mathcal{L}_{total}^{dis} = \lambda_{adv}\mathcal{L}_{adv}^{dis} \tag{19}$$

where $\lambda_{adv}$ and $\lambda_0$ are weights balancing the terms.

## 4 EXPERIMENTS

To evaluate the proposed method, we conduct experiments using a publicly available dataset [12]. As there are no prior studies exploring the end-to-end solutions in X-ray image synthesis from volume data, we benchmark our proposed method against the state-of-the-art (SOTA) 3D-aware GANs, including $\pi$-GAN [8] and EG3D [7]. The experimental settings are presented in Section 4.1. The evaluation of the proposed method and the ablation study are included in Section 4.2 and Section 4.3, respectively.

## 4.1 Experimental Settings

**Implementation Details**. For the CT encoder, we comprise an input layer, three downsampling layers and a latent layer. Each layer comprises two 3D convolutions (Conv3D) followed by Batch-Norm (BN) and activated by LeakyReLU with a slope of 0.2. After the latent layer of the CT encoder, a transformation layer flattens the output feature map into a 1D vector. AdaIN [16] is employed to form the synthesis layer of our generator. The style decoupling encoder $E_{sty}$ includes seven 2D convolution (Conv2D) layers in each of its two style branches. Before reaching the final layer, features are aggregated into a feature map through adaptive average pooling. The map is then passed through the final Con2D layer and activated by Tanh. The generator comprises a total of fourteen synthesis layers. These layers are sequentially numbered based on the resolution increasing of intermediate feature maps. Layers one to eight are designated as the content layers, primarily responsible for generating anatomical structures of the X-ray image. Conversely, the ninth to fourteenth layers, characterized by higher resolutions, serve as fine layers for injecting style information. The framework is trained with a batch size of 16 on a single NVIDIA RTX A6000 GPU with 48 GB of GPU memory. The proposed method utilizes

the Adam optimizer with a learning rate of 0.0025 and is trained for 100 epochs.

*Dataset*. Due to the lack of public datasets for our task, we construct the dataset to train our model. Specifically, public dataset CTSpline1K[12] includes 807 spine CT scans and is regarded as our CT data source. The scans are resampled to a resolution $1 \times 1 \times 1$ mm$^3$ and reshaped into $128 \times 128 \times 128$. Multi-view DRRs are generated from the CTSpine1K dataset. We utilize the DeepDRR [30] framework to generate DRRs. Considering the clinical emphasis on horizontal camera angles in C-arm imaging, we fix the camera angle at $0°$ vertically and project the CT from $-90°$ to $90°$ horizontally at intervals of $30°$ to create the DRR dataset. The parameters for DeepDRR are set as follows: step size of 0.1, spectral intensity of 60KV_AL35, photon count of $1,000,000$, source-to-object distance (SOD) of $1,020$ mm, and source-to-detector distance (SDD) of 530 mm. As for the X-ray images, a total of 373 real X-ray images are collected in the anterior-posterior (AP) and lateral (Lat) views from 186 patients.

*Baselines*. Currently, there are no available methods for the CT2X synthesis to serve as baselines. Hence, we selected SOTA 3D-aware image generation methods for comparison. $\pi$-GAN [8] generates multi-view images from view-consistent radiance fields based on volumetric rendering. EG3D [7] introduces a dedicated neural render for tri-plane hybrid 3D representation to generate high-quality multi-view images, enabling unsupervised multi-view synthesis. Both methods produce high-quality multi-view consistent images, which is particularly crucial for X-ray synthesis tasks [35]. These methods are not originally designed for the CT2X task and thus cannot directly process CT data. To bridge this gap and conduct a unfair comparison, we utilize the same CT encoder as our method to encode the CT scan into a latent code. The latent code can then be sent into $\pi$-GAN and EG3D and generate the X-ray results. The learning rate and number of epochs are also set to 0.0025 and 100, respectively.

*Evaluation Metrics*. We evaluated and compared the proposed method with regard to the metrics of image similarity and synthesis quality, respectively. For image similarity, we utilize LPIPS[47], which compares the semantic similarity between the generated images with reference images at a perceptual level through a pretrained deep network. For assessing the synthesis quality, we utilize the commonly used Frechet inception distance (FID) metric [14], which evaluates the distribution similarity by comparing reference and synthesized image distributions. Additionally, for evaluating the consistency with human visual perception, we employ the kernel inception distance (KID) metric [5] to assess the visual quality of synthesized images.

## 4.2 Experimental Results

*Qualitative Comparisons*. Figure 4 illustrates the qualitative comparison results with the baseline methods. From the figure, we can discern several noteworthy observations. Firstly, for image synthesis methods using 3D-aware networks, whether employing implicit-based ($\pi$-GAN) or hybrid-based (EG3D) 3D representations, difficulties arise in maintaining the accuracy of anatomical structures despite their ability to generate multi-view images.

Furthermore, both $\pi$-GAN and EG3D are unable to perform style-decoupling injection and can only utilize DRRs as target images to train the model, limiting them to generating results close to the DRR domain. In contrast, our method enhances feature information associated with the target pose in the content code through the incorporation of PAM. This enables multi-view image synthesis based on the corresponding view angle projection. Moreover, the style decoupling encoder extracts style features from X-ray images, facilitating the synthesis of results more closely resembling real X-ray images. In summary, our approach yields superior quality and clearer anatomical structures, making them more time-efficient and clinically applicable.

**Table 1: Quantitative comparison with the state-of-the-art synthesis methods.**

| Method | FID ↓ | KID ↓ | LPIPS ↓ |
|---|---|---|---|
| $\pi$-GAN | 277.3511 | 0.3131 | 0.4681 |
| EG3D | 224.3884 | 0.2535 | 0.2970 |
| Ours | **97.8350** | **0.0842** | **0.2366** |

*Quantitative Comparisons*. Table 1 presents the FID, KID and LPIPS values of our method and the SOTA methods, respectively. As can be seen from the table, the proposed method achieves the highest FID and KID scores, demonstrating that CT2X-GAN effectively captures the anatomical structure from the 3D CT scan and utilizes it to synthesize high-quality X-ray images. Besides, our method achieves the highest LPIPS for perceptual-level evaluation. This indicates that CT2X-GAN can successfully inject the style information of real X-ray images, corroborating the practicality of our approach. In summary, CT2X-GAN both achieves high anatomical structure fidelity and is realistic of the synthesized X-ray images.

## 4.3 Ablation Study

To understand the role of each component in CT2X-GAN, we conduct a series of ablation studies. Table 2 presents the quantitative results under all configurations, while Figure 5 illustrates the visualization examples of the outcomes.

**Table 2: Quantitative evaluation of ablation studies.**

| Method | FID ↓ | KID ↓ | LPIPS ↓ |
|---|---|---|---|
| w/o $E_{sty}$ | 160.0366 | 0.1736 | 0.3678 |
| w/o PAM | 156.5380 | 0.1715 | 0.3371 |
| w/o $\mathcal{L}_{consis}$ | 141.6687 | 0.1419 | 0.2966 |
| Ours | **97.8350** | **0.0842** | **0.2366** |

*Effect of Style Decoupling Encoder*. Referring to the first row of Table 2 and the first column of Figure 5, we verify the effectiveness of the style decoupling encoder. Removal of the decoupling encoder results in generated images resembling DRR with significant differences from X-ray images. Additionally, by comparing the first and last columns of Figure 5, it is evident that the style decoupling encoder introduces style disentanglement, enabling the

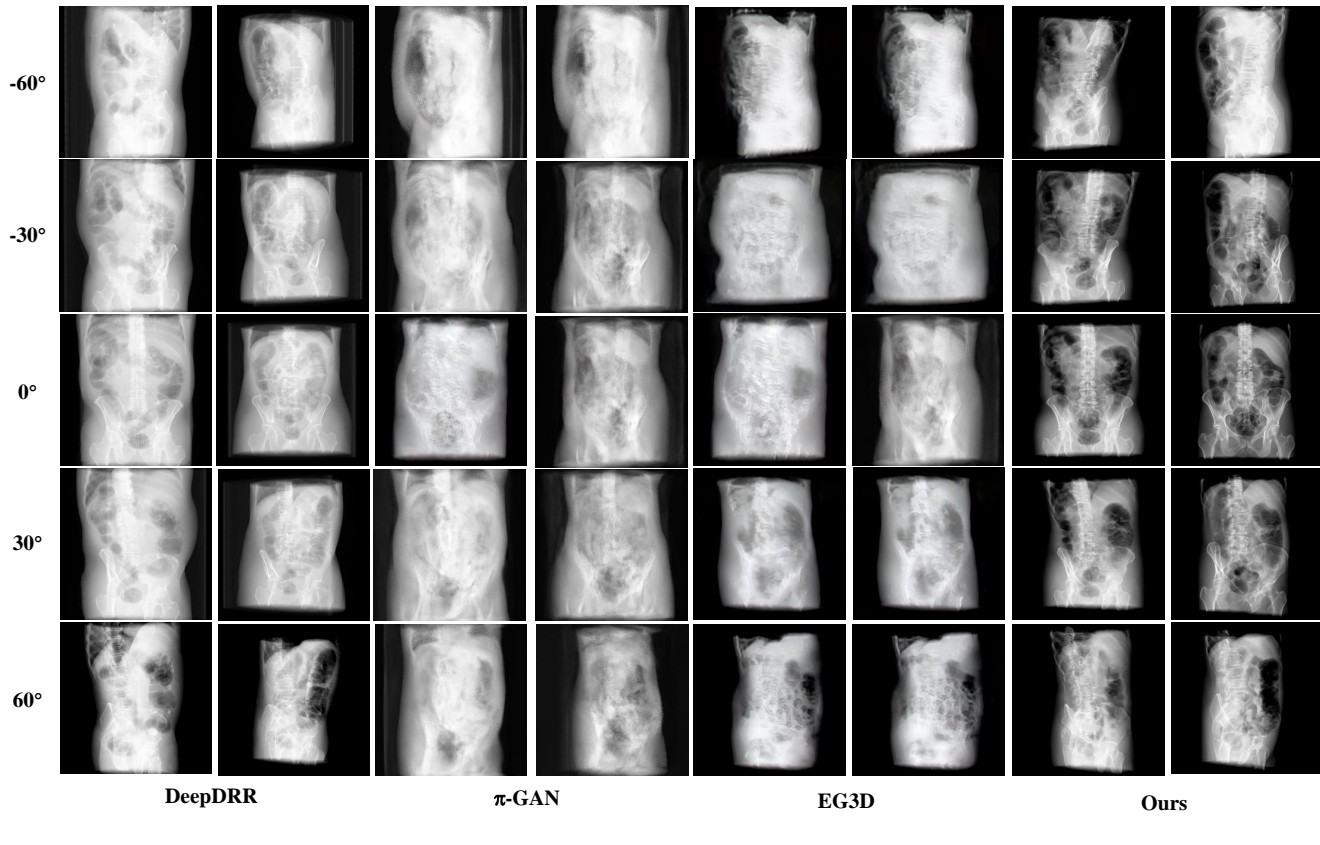

**Figure 4: Qualitative comparison between $\pi$-GAN, EG3D and ours at a resolution of** $256 * 256$.

generator to combine structural information from CT scans with style information from reference X-ray images. This can effectively make the reduced internal brightness and enhanced contrast in the skeletal parts of the generated results, as shown in the last column of Figure 5.

***Effect of Pose Attention Module***. From the second row of Table 2, we validate the effectiveness of PAM. It can be found that introducing PAM enhances the network awareness of pose information, resulting in better-synthesized X-ray images from multiple view angles. Results in Figure 5 show that the boundaries between skeletal and soft tissue parts become blurry. This indicates that PAM not only preserves the 3D spatial information in CT scans but also further pays more attention to anatomical structure information during the encoding process, further enhancing the quality of synthesized X-ray images.

***Effect of Consistency Regularization***. In this ablation study, we validate the effectiveness of the consistency regularization term. As shown in the third row of Table 2, removing style and content consistency regularization during training greatly harms the synthesis ability of our CT2X-GAN. The right two columns of Figure 5 demonstrate the results before and after introducing consistency regularization. It can be concluded the introduction of consistency regularization during training ensures the decoupling encoder. It

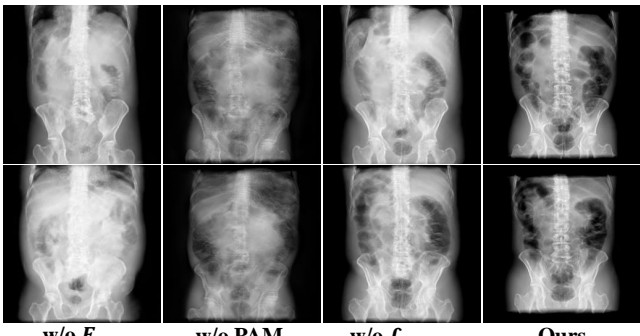

**Figure 5: Qualitative ablation study for proposed modules.**

avoids encoding anatomical information from the style reference images into the style code without affecting the final synthesis quality. The results indicate the ability of our proposed method to inject style while preserving anatomical structures.

## 4.4 Style Disentanglement Evaluation

***X-ray Style Synthesis***. To confirm the decoupling ability of the proposed style encoder, we evaluate the results given different style reference images for the same input CT scan. As depicted in

Figure 6, using style images from the same domain yields similar results, while using style images from different domains notably affects the style and intensity of the synthesized results.

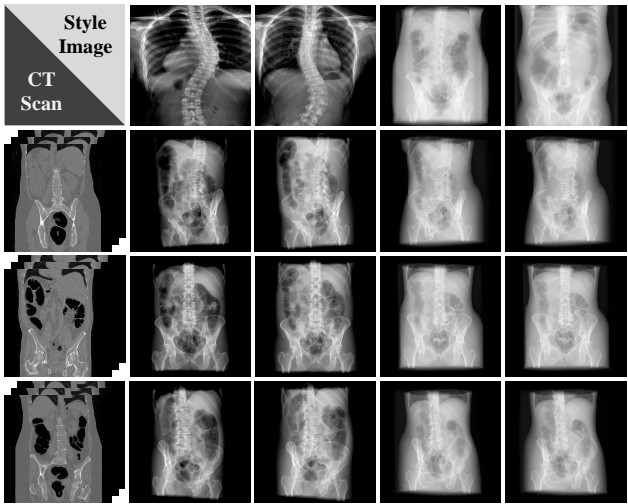

**Figure 6: Visual results for disentanglement module. From left to right: the different stylized results of the same input CT. From top to bottom: the results using different CT scans and the same reference style image.**

*DRR Style Reconstruction*. To quantitatively assess the style disentanglement capability of CT2X-GAN, we extract the DRR style information and incorporate it into the generator. The CT2X-GAN can generate high-quality reconstructed DRR images, as shown in Figure 7. In terms of quantitative metrics, we opted for structural similarity index measure (SSIM) and peak signal-to-noise ratio (PSNR). SSIM is employed to assess the structural accuracy of generated results in a manner of perceptual-level comparison, while PSNR provides a measure of the synthesis quality. It is noted that $\pi$-GAN and EG3D do not have the capability for style injection and are trained directly on DRR images. In such a way, their predictions are deemed suitable for use solely as reconstruction results. The quantitative results are presented in Table 3. It can be seen that our method outperforms others in all metrics, encompassing both SSIM and PSNR. This suggests that style disentanglement not only achieves stylized synthesis but also improves the network representation of anatomical information in CT data, reflecting anatomical structures more accurately.

**Table 3: Quantitative evaluation results for DRR style reconstruction.**

| Method | PSNR ↑ | SSIM ↑ |
| --- | --- | --- |
| $\pi$-GAN | 11.6527 | 0.2251 |
| EG3D | 15.0620 | 0.5046 |
| Ours | **23.5093** | **0.7013** |

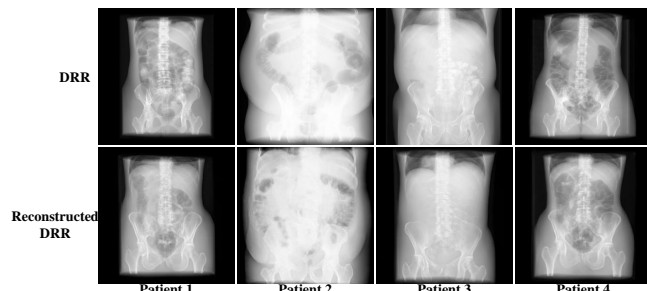

**Figure 7: Visual results for DRR style reconstruction.**

**Table 4: Results of user studies on the synthesis results of different models. The X-ray similarity indicates the resemblance of the synthesis obtained by each method to real X-ray images.**

| Method | DRR | $\pi$-GAN | EG3D | Ours |
| --- | --- | --- | --- | --- |
| X-ray Similarity | 2.6403 | 1.6406 | 1.7656 | **3.0938** |

## 4.5 User Study

We run a user study by randomly selecting 20 instances from the synthesized images and inviting 47 users to score them. Specifically, the participants are required to rate the synthesized images produced by various methods on a scale of 1-5, where higher scores signify greater resemblance to actual X-ray images. Subsequently, we compute the average score as the X-ray image similarity measure. The results are depicted in Table 4. The table reveals that the images synthesized by $\pi$-GAN and EG3D are notably distorted, as users predominantly perceive them to deviate substantially from authentic X-ray data. While superior to the 3D-aware methods, the DRRs still fall short compared to our method, which further demonstrates that our approach yields results more closely resembling to real X-ray images.

## 5 DISCUSSION

*Conclusion*. This study introduces a baseline method for multi-view X-ray image synthesis from CT scans. Our objective is to train a model that integrates content information with style features from unpaired CT and X-ray data in an end-to-end manner. This is accomplished by employing a novel decoupling learning method and consistency constraints. In addition, our method encourages the model to fully exploit the abundant spatial information included in CT scans through the PAM module. Compared to existing multi-view synthesis methods including $\pi$-GAN and EG3D, our approach can obtain multi-view X-ray results much more realistic.

*Limitations and future work*. Considering our method endeavours to tackle the challenge of limited paired data through disentanglement, it is inevitable leading to the sacrifice of anatomical information. Additionally, our method lacks distance awareness, resulting in scale distortion. To enhance the versatility and applicability of the synthesis process across diverse scenarios, investigating methods to perceive distances is needed and also will be our future work.

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
