# OpenReview forum: "Multi-view X-ray Image Synthesis with Multiple Domain Disentanglement from CT Scans"
_acmmm.org/ACMMM/2024/Conference — MM2024 Poster_

### Official Review · Reviewer_5oq3 · 2024-05-17

**Rating:** 2
**Confidence:** 3

**Summary:**

The paper present CT2X-GAN, a network to end-to-end synthesize the X-ray images from CT scans volume. It relieve the difficulty of collecting paired CT scans and X-ray images and it does not need to model the whole X-ray imaging procedure. The paper demonstrate CT2X-GAN outperforms the SOTA methods on the publicly available CTSpine1K dataset in terms of FID, KID, LPIPS.

**Strengths:**

- The proposed network enable end-to-end synthesize the X-ray images from CT scans volume.
- SOTA is achieved on the publicly available CTSpine1K dataset.
- The paper is generally well written.

**Limitations:**

- The author mentioned over-dosed X-rays superimpose potential risks to human health，so they designed CT2X-GAN to generate X-ray images from CT scans to avoide over-dosed X-rays. However, the common way which the Doctor choose to avoide over-dosed X-rays for patients is determine whether they need to take a CT scan or just X-ray image. Once the X-ray image can provide the doctor with enough information for diagnosis, the doctor will not ask the patient to take a CT scan. If the doctor requires the patient to take a CT scan, it means that the X-ray image is not enough to provide the information required for diagnosis, such as three-dimensional spatial information. At this time, the patient will be asked to directly take a CT scan, and the patient will not be asked to take an X-ray film on the basis of the CT scan. I hope the author will add more information about the necessity of generating X-ray images from CT scan to improve the significance of this study.
Furthermore, the CT volume input by the network in this article is actually three-dimensional reconstructed data. It seems more straightforward to directly perform image synthesis based on the original multi-angle two-dimensional projection of CT scan. The author's method of obtaining projections from reconstructed CT volume seems both inefficient and prone to introducing cumulative errors.

- In Table 1, The FID,KID and LPIPS evaluation metrics are all based on pure image evaluation. However, the most critical evaluation in X-ray imaging is that the generated X-ray image cannot draw wrong conclusions in the downstream diagnosis and other task evaluations. Can the authors use some diagnostic methods based on X-ray image to determine whether the CT2X-GAN generated X-ray image is better?
Some of the traditional methods of building physical models mentioned above are not compared in terms of evaluation metrics like FID and KID in this article. Can they be added in the future?

**Suitability:**

2

---

### Official Review · Reviewer_YYB3 · 2024-05-19

**Rating:** 6
**Confidence:** 3

**Summary:**

This work proposes a novel method for synthesizing multi-view X-ray images from CT scans. To overcome the challenge of preparing the paired CT and X-ray data to learn the mapping between these two imaging modules, the authors propose fusing the information from multiple domains, such as the style decoupling, CT encoder, and camera pose attention module. The consistency regularization term is introduced to constrain the generated data's content information and stylistic features. The pose attention module is designed to exploit the abundant spatial information in CT scans. Numerical results indicate that the newly proposed model produces high-quality synthesized X-ray images.

**Strengths:**

- The CT2X-GAN model poses a generative adversarial network (GAN) framework for multiple domain information disentanglement, which effectively solves the style injection problem in CT-to-X-ray image synthesis.
- The quantitative evaluation of the DRR style reconstruction revealed a remarkable improvement in PSNR, a result that is impress.
- The CT2X-GAN shows higher quality and clearer anatomical structures in the visualization results of multi-view image synthesis.

**Limitations:**

- The ribs in the generated images of Figure 7 do not appear consistent with those in authentic X-ray images.
- The technical details, particularly the implementation of the disentanglement learning and consistency constraints, are necessary.
- Further clarification on choosing the weight parameters in the hybrid loss function is needed.

**Suitability:**

3

---

### Official Review · Reviewer_rYhF · 2024-06-10

**Rating:** 4
**Confidence:** 2

**Summary:**

This manuscript focused on the task of X-ray image synthesis. In this work, the authors proposed, CT2X-GAN, an end-to-end framework to synthesize the X-ray images using the content and style disentanglement from three different image domains. Specifically, the anatomical structure (content) information and the style information were decoupled from the CT scans and the unpaired real X-ray/DRR images, respectively. For the style information, a consistency regularizer was proposed to improve the stylistic resemblance between real and synthesized X-ray images. Besides, zero loss was imposed to compute the similarity of real and synthesized DRR images. As for the content information, a pose attention module (PAM) was developed to enhance the decoupled content code from CT scans and facilitate multi-view image generation. Experiments on CT-Spine1K dataset were conducted to demonstrate the effectiveness of the proposed method, especially the superior about synthesis quality and realisticness.

**Strengths:**

1. The proposed method provided an end-to-end synthesis framework for X-ray images.
2. It was claimed that this manuscript was the first work that considered the CT-to-Xray synthesis.
3. Pose information was considered in this work to multi-view syntheses, which can enrich the diversity of the generated data and is valuable for real-world scenarios.

**Limitations:**

1. The overall framework seems to consist of many sub-modules with different loss functions. I wonder if it is easy to determine the coefficients of different loss components. However, there is no discussion about the sensitivity of these hyperparameters in the main paper or in the appendix.
2. This work focused on the multi-view synthesis of X-ray images from CT scans. Intuitively, the difficulties of the synthesis under different views/poses can be different. However, there is a lack of detailed quantitative results under different views/poses.

### Further comments/questions:
1. In Section 3.3, it is confusing if the style decoupling encoder $E_{sty}$ shared parameters when dealing with X-ray and DRR images. Intuitively, the decoupling and synthesis of DRR images were leveraged to train the style feature extractor on X-ray images. Based on this, it seems that the style branches for DDR images and X-ray images should share the same parameters (i.e., using the same branch $E_{sty}^{s_{DRR}}=E_{sty}^{s_{X}}$). However, from Eq(2)(3)(5)(6), it seems that they shared the content branch (i.e., $E_{sty}^{c_{DDR}}=E_{sty}^{c_{X}}=E_{sty}^{c}$) rather than the style encoder (i.e, $E_{sty}^{s_{DRR}}\neq E_{sty}^{s_{X}}$). The data stream within the style encoder $E_{sty}$ in Figure 2 is confusing. Could the authors provide a detailed explanation?

2. The math formulations should be improved to keep the consistency for reading experiences. For example, the style code of DDR images was introduced as $f_{sty}^{DDR}$ but it seems it was referred to as $f_s^{DRR}$. Please specify the right one if I misunderstand it.

3.  In Eq.(10), why was the content consistency $L_{cc}$ applied on the reconstructed images $\hat{I}_{X}$ and $\hat{I}_{DRR}$, rather than the original images  $I_{X}$ and $I_{DRR}$? Could the authors explain the motivation for such a design?

4. From Section 3, I wonder if the proposed methods is training-efficient compared to the other baselines. For example, in Eq. (11), it seems that the generated images $\hat{I}_{X}$ and $\hat{I}_{DRR}$ will be sent to the style encoder again for the loss computation. It would be better if the authors could provide a quantitative comparison with respect to the computational overhead.

5. The synthesis performance under different views can be a crucial criterion to evaluate the generation ability. I wonder if the synthesis difficulty for different views/poses (e.g., the degrees of -60, -30, 0, 30, and 60 considered in this manuscript) is identical. If not, it would be better if the authors could provide detailed quantitative results under different views/poses. I believe it is important to investigate the pros/cons of the proposed method and the baselines.

**Suitability:**

2

---

### Official Review · Reviewer_4ZNG · 2024-06-10

**Rating:** 5
**Confidence:** 4

**Summary:**

The paper proposes a novel approach called CT2X-GAN for synthesizing X-ray images from CT scans. The method aims to address the challenge of limited paired data through disentanglement, enabling the synthesis of high-quality and realistic X-ray images from multiple view angles. The authors introduce a CT encoder, a style decoupling encoder, a pose attention module, and a consistency regularization term to improve the synthesis process. The proposed method is compared with state-of-the-art 3D-aware methods, demonstrating superior performance in terms of FID, KID, LPIPS, SSIM, PSNR, and user-rated X-ray similarity.

**Strengths:**

The paper introduces a novel and practical task of CT-to-X-ray (CT2X) synthesis, offering an end-to-end approach without modeling the entire X-ray imaging procedure.
The proposed style decoupling encoder effectively extracts style features from real X-ray images, relieving the difficulty of collecting paired CT scans and X-ray images.
The novel regularization method utilizing consistency and zero loss improves style accuracy and structural consistency in multi-views, enhancing the synthesis quality.
The pose attention module enables the network to improve the perception ability of structural content at multiple view angles, facilitating high-quality multi-view image synthesis.

**Limitations:**

-The method sacrifices anatomical information due to the focus on disentanglement, leading to scale distortion and limitations in perceiving distances.
-The paper lacks a comprehensive discussion on the computational complexity and potential challenges in real-world implementation.

**Suitability:**

2

---

### Meta-Review · Area_Chair_tA3U · 2024-07-07

**Recommendation:** Accept (Poster)
**Confidence:** 4

**Metareview:**

Most of the reviewers vote for acceptance, and I suggest the authors carefully addressing some concerns (such as motivation and ablation studies) raised by the reviewers in the final version.